# Predicting gene regulatory regions with a convolutional neural network for processing double-strand genome sequence information

**Koh Onimaru**[ID]*, **Osamu Nishimura**[ID], **Shigehiro Kuraku**

Laboratory for Phyloinformatics, RIKEN Center for Biosystems Dynamics Research (BDR), Chuo-ku, Kobe, Hyogo, Japan

* koh.onimaru@riken.jp

## Abstract

With advances in sequencing technology, a vast amount of genomic sequence information has become available. However, annotating biological functions particularly of non-protein-coding regions in genome sequences without experiments is still a challenging task. Recently deep learning–based methods were shown to have the ability to predict gene regulatory regions from genome sequences, promising to aid the interpretation of genomic sequence data. Here, we report an improvement of the prediction accuracy for gene regulatory regions by using the design of convolution layers that efficiently process genomic sequence information, and developed a software, DeepGMAP, to train and compare different deep learning–based models (https://github.com/koonimaru/DeepGMAP). First, we demonstrate that our convolution layers, termed forward- and reverse-sequence scan (FRSS) layers, integrate both forward and reverse strand information, and enhance the power to predict gene regulatory regions. Second, we assessed previous studies and identified problems associated with data structures that caused overfitting. Finally, we introduce visualization methods to examine what the program learned. Together, our FRSS layers improve the prediction accuracy for gene regulatory regions.

## Introduction

In the last decade, advances in DNA sequencing technology have dramatically increased the amount of genome sequence data derived from diverse species [1] as well as from individual humans [2]. The next demanding challenge is the deeper understanding of how genome sequences encode phenotypes and how functional information can be extracted [3]. Such sequence-based understanding would ultimately enable the prediction of phenotypes based on genome sequence information, i.e., genotype-phenotype mapping. The syntax of protein-coding genes is well understood, e.g., certain phenotypic consequences are predictable (such as nonsense mutations), yet the basic rules for non-coding sequences have not been established. Several projects including ENCODE [4,5], ROADMAP [6], and FANTOM [7] have accumulated epigenomic data to annotate the characteristics of non-coding sequences, but a

**Data Availability Statement:** All codes used in this paper are available at https://github.com/koonimaru/DeepGMAP. The IDs for data downloaded from the ENCODE website are listed in S2 Table. The data generated and/or analyzed in the current study are available in the figshare repository, https://doi.org/10.6084/m9.figshare.6728348.

**Funding:** This work was supported in part by JSPS KAKENHI grant number 17K15132 to KO, a Special Postdoctoral Researcher Program of RIKEN to KO,

and a research grant from MEXT to the RIKEN Center for Life Science Technologies and RIKEN Center for Biosystems Dynamics Research.

**Competing interests:** The authors have declared that no competing interests exist.

comprehensive understanding of the association between epigenomic signatures and sequence information has yet to be attained.

Recently, this challenge has been addressed with deep learning–based methods [8–11]. Deep learning is a subfield of machine-learning methods, and has been applied to a variety of problems such as image classification and speech recognition [12]. For the application of deep learning to genomics, convolutional neural networks (CNN) or recurrent neural networks (RNN) are trained with input genomic sequences that are labeled with epigenomic data to predict functional non-coding sequences. Once trained, these types of classifiers can infer the functional effect of mutations in non-coding genomic regions from individual genome sequences. However, even though these deep learning–based classifiers have outperformed other methods, such as the support vector machine [13], the prediction accuracy is still far from satisfactory. In addition, as it is generally known [14], it remains elusive as to what deep learning–based models actually "learn" and what kinds of "understanding" underlies their predictions. In the present study, we first describe a CNN-based classifier that can outperform state-of-art models. Second, we identified problems associated with data structures that caused overfitting. Finally, we introduce methods to visualize trained models, potentially revealing the general syntax underlying regulatory sequences.

## Results

To improve the accuracy of predicting regulatory sequences, we devised a deep learning–based method with three main features: a) integrating information from forward and reverse DNA sequences; b) simplifying the data structure; c) quality control to filter out low-quality data from a training dataset.

### Training design and benchmarking

As training data, we downloaded several alignment files containing data from chromatin accessibility assays and chromatin immunoprecipitation-sequencing (ChIP-seq) experiments from the ENCODE project, and regions enriched with reads were determined as peaks by MACS2 peak caller [15]. These data were used to mark genome sequences. We divided each of the mouse and human genome sequences into 1000-basepair (bp) windows and converted the four letters (i.e., A, C, G, T) into one-hot vectors (four dimensional zero-one vectors). To denote epigenomic marks, we assigned each window as 1 if it overlapped a signal positive region or 0 otherwise (signal negative) (see Methods for details). To reduce the number of potential artifacts caused by window boundaries, we also added windows that were shifted by 500 bp toward the 3′ side (Fig 1A; we refer to this window structure as 1-kbp window/0.5-kbp stride). These data were used to train CNN models (see S1 Fig for the overall scheme).

To find an effective architecture of neural networks, we first compared the network architecture of published models such as CNN-based models (DeepSEA [9] and Basset [16]) and a CNN-RNN hybrid (DanQ [10]; see S1 Table for details of the models). Because these models were implemented in different programming languages, we re-implemented them in our code (referred to as the DeepSEA-type model and so on) to examine only the effect of network architectures. In addition, because the aim of this benchmark is to find an architecture that can be efficiently trained with limited data (all epigenomic data are fundamentally limited by the genome size and the binding frequency of transcription factors), and the number of training epochs was constrained to one, which means that models were trained with the full dataset only once. Moreover, because full training of models sometimes take days or weeks, we simplified the training task in this benchmarking as follows: only a subset of mouse DNase-seq data was used (the data IDs are listed in S2 Table). As a result, the training dataset for this

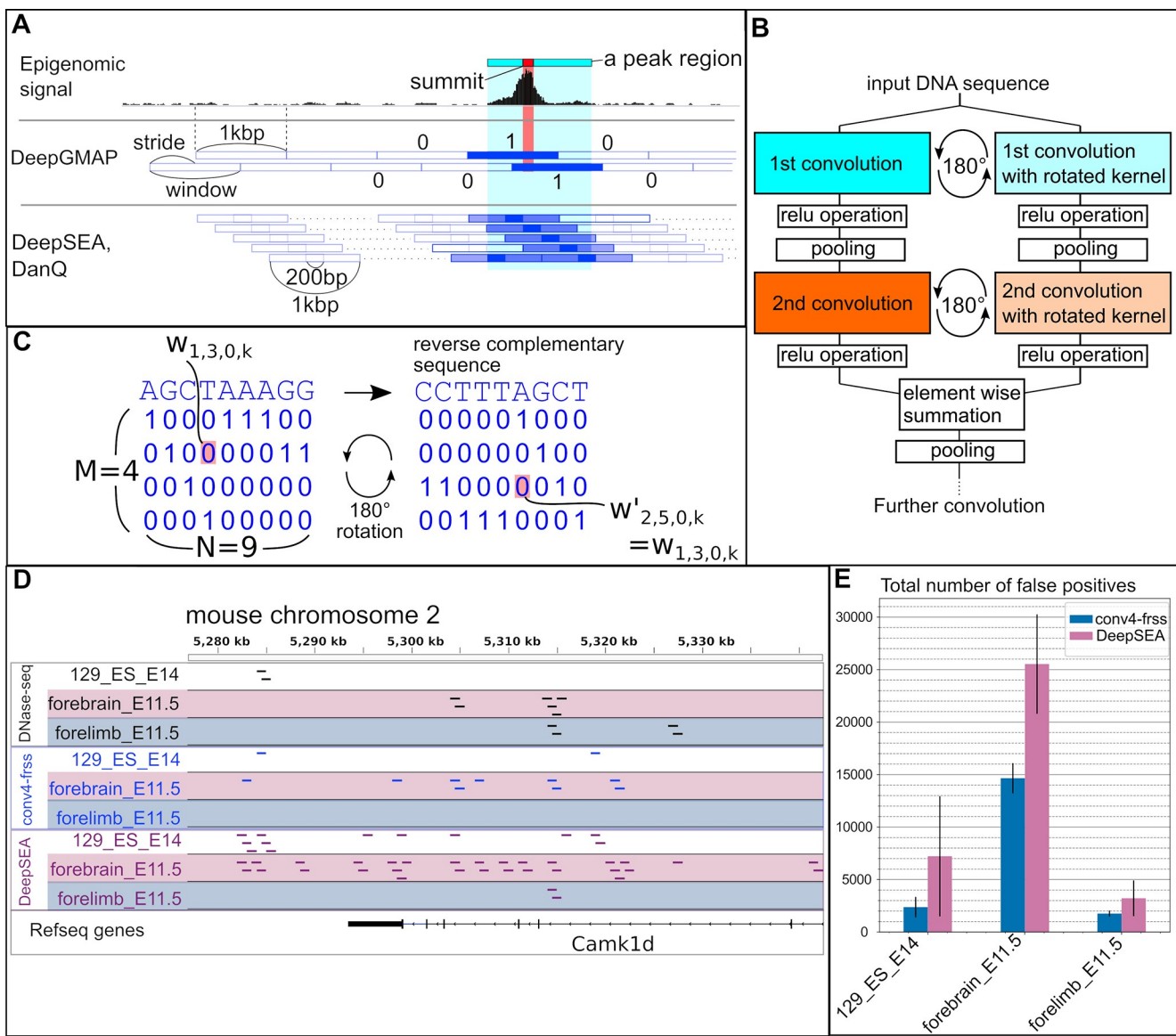

**Fig 1. The Forward- and Reverse-Sequence Scan (FRSS) layers and data structures.** (**A**) Schematic representation of how to assign an epigenomic signal into a binary vector. A peak region (light blue bar) and the summit of a peak (red rectangle) are determined from epigenomic data and assigned as "1" to overlapping genomic regions (blue-filled rectangle), otherwise "0" (blue, empty rectangle). DeepGMAP, this study; DeepSEA, DanQ, previous studies. (**B**) The architecture of the FRSS layers. An input sequence is scanned by the first convolution kernels, and the kernels are rotated in parallel. After implementing the relu operation, pooling, and the second convolution, the two parallel outputs are combined through summation. (**C**) Illustration of the 180-degree rotation of kernels. Each variable in an M × N kernel is written as $w_{m,n,h,k}$ ($[m] = \{0,\ldots,M{-}1\};[n] = \{0,\ldots,N{-}1\};[h] = \{0,\ldots,H{-}1\};[k] = \{0,\ldots,K{-}1\}$; H and K are the numbers of the channels and kernels, respectively). $w'_{m',n',h,k}$ is a variable after the 180-degree rotation, and equal to $w_{(M{-}1{-}m),(N{-}1{-}n),h,k}$ (also see Methods). In this example, a 9 × 4 kernel is trained to detect a sequence, AGCTAAAG (left). The geometric rotation of the kernel (right) results in the reverse complement of the original sequence. (**D**) A visual comparison between DNase-seq peaks and predictions of the conv4-FRSS and DeepSEA models. Black boxes indicate 1 kb-windows that overlap with peaks. Blue and purple boxes indicate peak regions predicted by the indicated models. Note that DeepSEA predictions contains many false positives. (**E**) The total number of false positives predicted by the DeepSEA model (purple) and the conv4-FRSS model (blue). The averages were calculated from predictions by three independently trained models.

benchmark is composed of 16747 mini-batches (each mini-batch contains 100 data). For evaluation, we used the entire chromosome 2 sequence, which was excluded from training datasets, and calculated the prediction accuracy of models using two scores, namely the area under the receiver-operation curve (AUROC) and the area under the precision-recall curve

**Table 1. A performance comparison between models.**

| | 129_ES_E14 | | forebrain E11.5 | | forelimb E11.5 | | mean | | time[b] |
|---|---|---|---|---|---|---|---|---|---|
| | **AUROC** | **AUPRC** | **AUROC** | **AUPRC** | **AUROC** | **AUPRC** | **AUROC** | **AUPRC** | |
| DeepSEA-type | 0.9234 | 0.6038 | 0.9141 | 0.6380 | 0.9315 | 0.5734 | 0.9230 | 0.6051 | 0.1773 |
| Basset-type | 0.9085 | 0.5796 | 0.9107 | 0.6280 | 0.9175 | 0.5476 | 0.9123 | 0.5851 | 0.0847 |
| DanQ-type | 0.9057 | 0.5785 | 0.9076 | 0.6214 | 0.9046 | 0.5164 | 0.9059 | 0.5721 | 0.4586 |
| DanQ-type block[a] | 0.9053 | 0.5699 | 0.9063 | 0.6168 | 0.9030 | 0.5031 | 0.9049 | 0.5633 | 0.4832 |
| conv4 | 0.9300 | 0.6234 | 0.9146 | 0.6426 | 0.9351 | 0.5884 | 0.9266 | 0.6181 | 0.2548 |
| conv3-FRSS | 0.9364 | 0.6447 | 0.9171 | 0.6496 | 0.9421 | 0.6098 | 0.9319 | 0.6347 | 0.3636 |
| conv4-FRSS | **0.9414** | **0.6571** | **0.9178** | **0.6520** | **0.9476** | **0.6280** | **0.9356** | **0.6457** | 0.3729 |
| conv4-FRSS+ | **0.9418** | **0.6631** | **0.9183** | **0.6549** | **0.9475** | **0.6347** | **0.9359** | **0.6509** | 0.5802 |
| conv4-nonFRSS | 0.9331 | 0.6424 | 0.9143 | 0.6407 | 0.9388 | 0.6008 | 0.9287 | 0.6280 | 0.3760 |
| DanQ-FRSS | 0.9365 | 0.6523 | 0.9162 | 0.6485 | 0.9453 | 0.6222 | 0.9327 | 0.6410 | 0.7905 |

[a]Tensorflow has several LSTM cells, and this model uses LSTMBlockCell, and the above uses LSTMCell. [b]the mean time (second) of 20 minibatch updates.

(AUPRC). Because AUROC is not a proper criterion if the majority of the data is negatively labeled, we considered AUPRC as a more important criterion [17]. As a result, we found that the DeepSEA-type architecture performed better than the Basset-type and DanQ-type models at least in this training condition (Table 1).

## Reading forward and reverse-complementary sequences improve prediction accuracy

Based on the above result, we next explored whether better models could be attained by modifying the DeepSEA architecture. First, as deeper convolutional networks are known to perform better [18], we inserted an additional convolution layer between the third convolution layer and the first fully connected layer of the DeepSEA-type model (with a reduced kernel number in some layers and smaller pooling patches). As expected, the four-layer convolutions achieved slightly better accuracy (conv4 in Table 1). We next focused on one of the fundamental characteristics of genome sequences—the forward and reverse strands. In previous studies, the information on reverse-strand sequences was ignored or used as independent data. However, for example, transcription factors with leucine-zipper or helix-loop-helix domains bind both strands through dimerization, and such binding thus results in palindromic binding motifs [19,20]. Inspired by this fact, we devised a set of layers that can integrate information from the both strands (Fig 1B and Methods). We arranged the one-hot vectors that represent AGCT in a symmetric manner so that a 180-degree rotation of the one-hot vectors results in a reverse complementary sequence (Fig 1C). Thus, reverse-sequence information can be processed by rotating the kernels 180 degrees. The hidden layers derived from the two strands are combined by element-wise addition after the second convolution. We termed this architecture FRSS (forward- and reverse-sequence scan) layers. The replacement of the first two convolutional layers in the DeepSEA-type model with FRSS (conv3-FRSS in Table 1) improved the accuracy of prediction more than the simple addition of a convolution layer (conv4). Moreover, a four-layer convolution model with FRSS (conv4-FRSS) achieved higher AUPRC scores than the other models (Table 1, and Fig 1D for a visual comparison). Although the magnitude of increase in AUPRC seemed subtle, the Fig 1D clearly showed a significant decrease in false positives. This visual intuition was validated by the total counts of false positives; the number of false positives by the DeepSEA-type model varied among replicates and was higher than those by the conv4-FRSS model (Fig 1E). Raising the kernel numbers of each convolution layer also slightly

improved the performance (conv4-FRSS+ in Table 1) but also increased computation time (the right most column in Table 1). To confirm that the effect of FRSS resulted not from merely increase in the kenerl number but from the rotation of kernels, we replaced the rotated kernels with independently trainable kernels in the conv4-FRSS (named as "conv4-nonFRSS" in Table 1). The replacement resulted in a poorer performance than the conv4-FRSS model, indicating that the rotated kernels improve training efficiency. We also found that the FRSS layers increased the learning efficiency of the DanQ model by a level comparable with conv4-FRSS (DanQ-FRSS in Table 1). These results were consistently reproduced by repeated training and testing (S3 Table), indicating that the benchmark is robust against random initialization. Together, these results suggest that the FRSS layers enhance the predictive power of deep learning–based models.

## The length of strides affect training efficiency

Next, we examined the structure and quality of training data using the conv4-FRSS model. In certain previous studies, epigenomic signals were distributed into 200-bp windows of genome sequences, and 400-bp extra-sequences were added to the left and right sides of each window [9] (referred to as "DeepSEA-type data" in this study; Fig 1A). For comparison, we trained conv4-FRSS with the DeepSEA-type data structure. Training with the DeepSEA-type data yielded a higher AUROC score but a lower AUPRC score (DeepSEA-type in Table 2). In addition, we tested several window and stride sizes and found that models can be trained most efficiently with a 1-kbp window and 0.3-kbp stride data. These results indicated that data-structure design is a critical factor for training models efficiently and that data augmentation using small strides, which was implemented in several studies [21,22], is not an optimal strategy for training models.

## conv4-FRSS outperforms previous models

To validate the performance of the FRSS layers, we extended the list of training data. First, we newly trained conv4-FRSS with the same set of 125 human DNase-seq data used by Zhou & Troyanskaya (2015) [9] (see S2 Table for the lists of data). We found that conv4-FRSS significantly outperformed DeepSEA and DanQ (Fig 2A and 2C and S4 Table; the scores of DeepSEA and DanQ were obtained from their publications; https://media.nature.com/original/nature-assets/nmeth/journal/v12/n10/extref/nmeth.3547-S3.xlsx for DeepSEA and auc.txt of https://github.com/uci-cbcl/DanQ for DanQ). We also trained conv4-FRSS with 365 and 93 of

**Table 2. A performance comparison between data structures.**

| Data structure | chromosome 2 (true test) | | chromosome 1 (overfitting test) | |
|---|---|---|---|---|
| | **AUROC** | **AUPRC** | **AUROC** | **AUPRC** |
| DeepSEA-type | 0.9623 | 0.4841 | 0.9694 | 0.4741 |
| 1 kb window/0.5 kb stride[a] | 0.9356 | 0.6457 | 0.9453 | 0.6293 |
| 1 kb window/0.4 kb stride | 0.9352 | 0.6528 | 0.9457 | 0.6386 |
| 1 kb window/0.3 kb stride | **0.9389** | **0.6570** | 0.9515 | 0.6618 |
| 1 kb window/0.2 kb stride | 0.9397 | 0.6530 | 0.9581 | 0.6962 |
| 1 kb window/0.1 kb stride | 0.9368 | 0.6396 | 0.9689 | 0.7757 |

The mean values of the three mouse DNase-seq data are shown. The columns under chromosome 2 are the results of true tests (the higher is better). The columns under chromosome 1 are overfitting tests (if the scores are higher than those of true tests, that is the sign of overfitting). [a]The same data with the mean values of conv4-FRSS in Table 1.

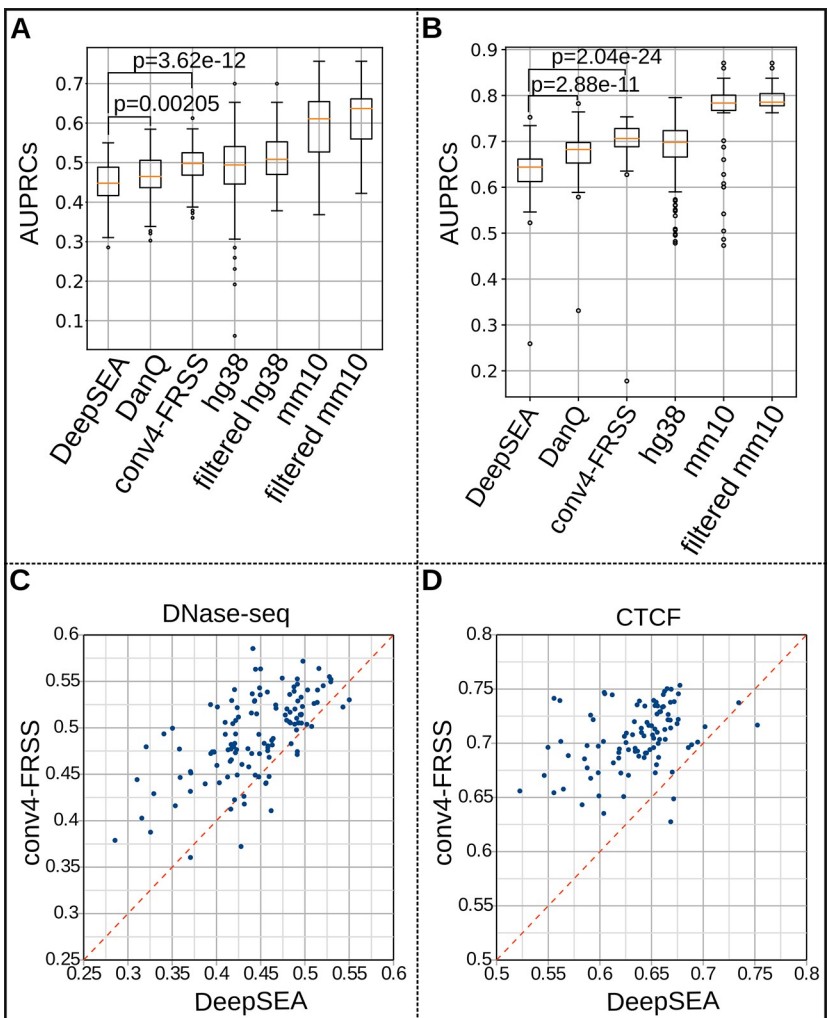

**Fig 2. Performance analyses of models with extended datasets.** (A, B) Comparison of AUPRCs of DNase-seq data (A) and CTCF data (B), respectively. The scores of DeepSEA and DanQ are obtained from their original studies. conv4-FRSS. hg38 and mm10 are the scores of conv4-FRSS with newly created training data using datasets obtained from the ENCODE project. filtered hg38 and mm10 are the prediction scores of conv4-FRSS trained with high-cFRiP datasets. p, two-sided Mann–Whitney–Wilcoxon test. Boxes, the lower to upper quantile values of the data; orange lines, the median; whiskers, the range of the data (Q1 –IQR × 1.5 and Q3 + IQR × 1.5 for the lower and upper bounds, respectively); flier points, those past the end of the whiskers. (C, D) Scatter plots of AUPRCs of DeepSEA (x axis; scores are obtained from its original data) and conv4-FRSS (y axis).

human and mouse DNase-seq peaks, respectively ("hg38" and "mm10" in the Fig 2A; see S2 Table for the lists of data). As shown in the Fig 2A and S4 Table, we noticed that training with mouse data yielded higher performance than with human data. We also focused on CTCF ChIP-seq data because (a) CTCF has a general function in genome organization, (b) ENCODE releases plenty of CTCF data for both mice and humans, and (c) as evident for the AUPRC scores of DeepSEA (S2 Fig; reanalyzed data from Zhou & Troyanskaya, 2015 [9]), the prediction accuracy is highly dependent on the target of ChIP-seq, and that of CTCF is the highest among transcription factors. As a result, the conv4-FRSS model was efficiently trained with both human and mouse CTCF data, and this model yielded a median AUPRC score of >0.70 (Fig 2B and 2D; the scores of DeepSEA and DanQ were obtained from their publications). Together, these results validate the performance of the FRSS.

## The quality control of training data

In the above training with extended dataset, we suspected that several classes with low AUPRCs were attributable to poor data quality. To address this possibility, we calculated a known quality index, FRiP (fraction of reads in peaks [23]), of mouse CTCF data and found that classes with low AUPRCs tended to have low FRiPs (top panels in the S3 Fig), with some exceptions (arrows in the S4 Fig and S5 Table). Because FRiP is calculated based on reads in peaks per total reads, values may be inflated when the read number of the source data is too low. To correct such inflation, we multiplied FRiP by the fraction of genomic regions covered by at least one read in a genome. As shown in the S3 Fig, the corrected FRiPs (cFRiPs) could distinguish high and low AUPRC data more clearly. In addition, cFRiP yielded a stronger correlation with the total peak numbers of data than the uncorrected FRiP (bottom panels in the S3 Fig). Although the other datasets showed weaker relations between FRiP and AUPRC, data with optimal cutoff values of cFRiP increased the average AUPRC score except for the human CTCF data (filtered hg38 and mm10 data in the Fig 2A and 2B, and S4 Fig). These results suggested two points. First, the correction method reasonably represents the true quality of the data. Second, the linear correlation between peak numbers and cFRiPs implies that data quality is not saturated enough to detect all true peaks (and hence, the possibility remains that the model learns sample quality rather than cell-specific patterns). Therefore, whereas the ENCODE project seems to consider ≥1% FRiP as good data [23], our results suggest that higher stringency during quality control is required for the precise annotation of genomes.

## The utility of conv4-FRSS

Because the conv4-FRSS model showed a good accuracy for CTCF binding site prediction, we further examined the utility of this model. A comparison between the predictions of the model and CTCF motif regions detected by a motif scanning tool, FIMO [24] (FIMO in the Fig 3A) revealed that the model can predict CTCF binding regions that do not contain the canonical CTCF motif. In addition, we also used the class saliency extraction method [25] to identify informative sequences at the single-nucleotide level, which produce data similar to ChIP-seq signals (influential value in the Fig 3A and S5 Fig). This method is equivalent to the *in silico* saturation mutagenesis approach used previously [8,9] but is more computationally efficient because it does not require mutation-by-mutation evaluation (see Methods for details). Furthermore, using the dog genome (CanFam3.1), we performed a cross-species prediction of CTCF binding regions with the model trained solely with mouse data. As shown in the Fig 3B, the predictions of the model matched with the real CTCF signals of dog liver [26] and detected species-specific differences (compare red rectangles in the Fig 3), indicating the generality of the model. Based on all these data, we concluded that conv4-FRSS can be applied to diverse data.

## Visualization of what deep-learning models learn about genome sequences

To obtain insights into how the models predict regulatory sequences, we visualized conv4-FRSS trained with the subset of mouse DNase-seq data. First, as has been done in previous studies [8,10], we analyzed individual kernels of the first layer, which directly interact with DNA sequences and thus represent DNA motifs that are important for classifying regulatory sequences. We converted the weight variables to probability matrices by using a softmax-like operation and found that established models have indeed learned many known transcription factor binding motifs, such as those bound by CTCF, SOX9, OCT4 and KLF4 (Fig 4A, S6 Fig and see S6 Table for full comparison between the kernels and known motifs). Next, we applied the activation maximization method [25,27,28], in which we look for DNA sequences that

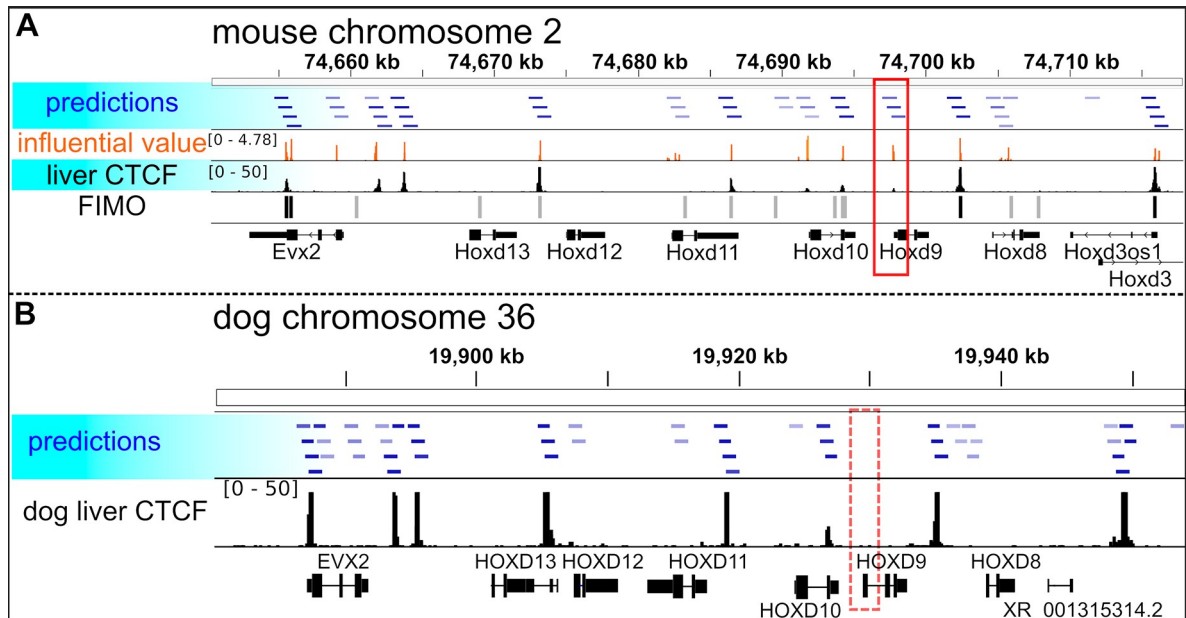

**Fig 3. An example of CTCF binding site predictions on the HoxD cluster of mice and dogs.** (**A**) Comparison between predictions by conv4-FRSS, experimaental data and motifs detected by FIMO in mouse chromosome 2 sequence. predictions, CTCF binding regions predicted by conv4-FRSS (prediction values ≥0.2 are shown, and the bluer is more probable); influential value, values calculated with the class saliency extraction method; mouse liver CTCF, alignment data from ENCFF627DYN; FIMO, CTCF sites that were detected by FIMO (black boxes, p ≤ 1e-4; gray boxes, 1e-4 < p ≤ 1e-3). (**B**) Comparison between predictions by conv4-FRSS and experimaental data in the sequence of dog chromosome 36. dog liver CTCF, alignment data from ERR022304; red rectangles, a CTCF peak that is present in the mouse genome (solid rectangle) but not the dog genome (dashed rectangle).

maximize neuron activities in the final layer of a trained model by training the sequence itself (see Methods for formal mathematical expressions). First, we trained a sequence to activate the 129 ES E14–specific neuron in the last output layer. As with image-recognition studies [25,28], this method generated randomly repeated sequences that probably capture the 129 ES E14 class-specific traits (Fig 4B). Using the motif comparison software Tomtom [29], we found that the generated sequence contained the motifs of several reprogramming-related transcription factors, such as OCT4/SOX2 and KLF4, which are important for the pluripotency of embryonic stem cells [30] (the full detected motifs are listed in S7 Table). For comparison, we also trained a sequence to activate all of the three classes of the neurons, resulting in a CTCF motif–rich sequence (Fig 4C). This result is consistent with the general role of CTCF as an enhancer looping factor [31]. In addition, the generated sequence also contained small motifs next to the well-known 20-bp motif (dashed boxes in the Fig 4C, which resembled the other part of the alternative long CTCF motif (33/34 bp in length, including an additional GGNANTGCA or TGCANTNCC sequence) [26] (the full detected motifs are listed in S8 Table). Thus, the detection of motifs that are longer than the kernel size constitutes one of the advantages of this activation maximization method. Taken together, the activation maximization method helps understand how deep-learning models predict enhancer sequences.

## Discussions

Our results demonstrate that a new convolution architecture, FRSS, efficiently discerns patterns of regulatory DNA sequences. Furthermore, the analyses of data quality and structure revealed that training efficiencies highly depend on the means by which the source data are filtered and processed. Although the visualization methods are still in development, these

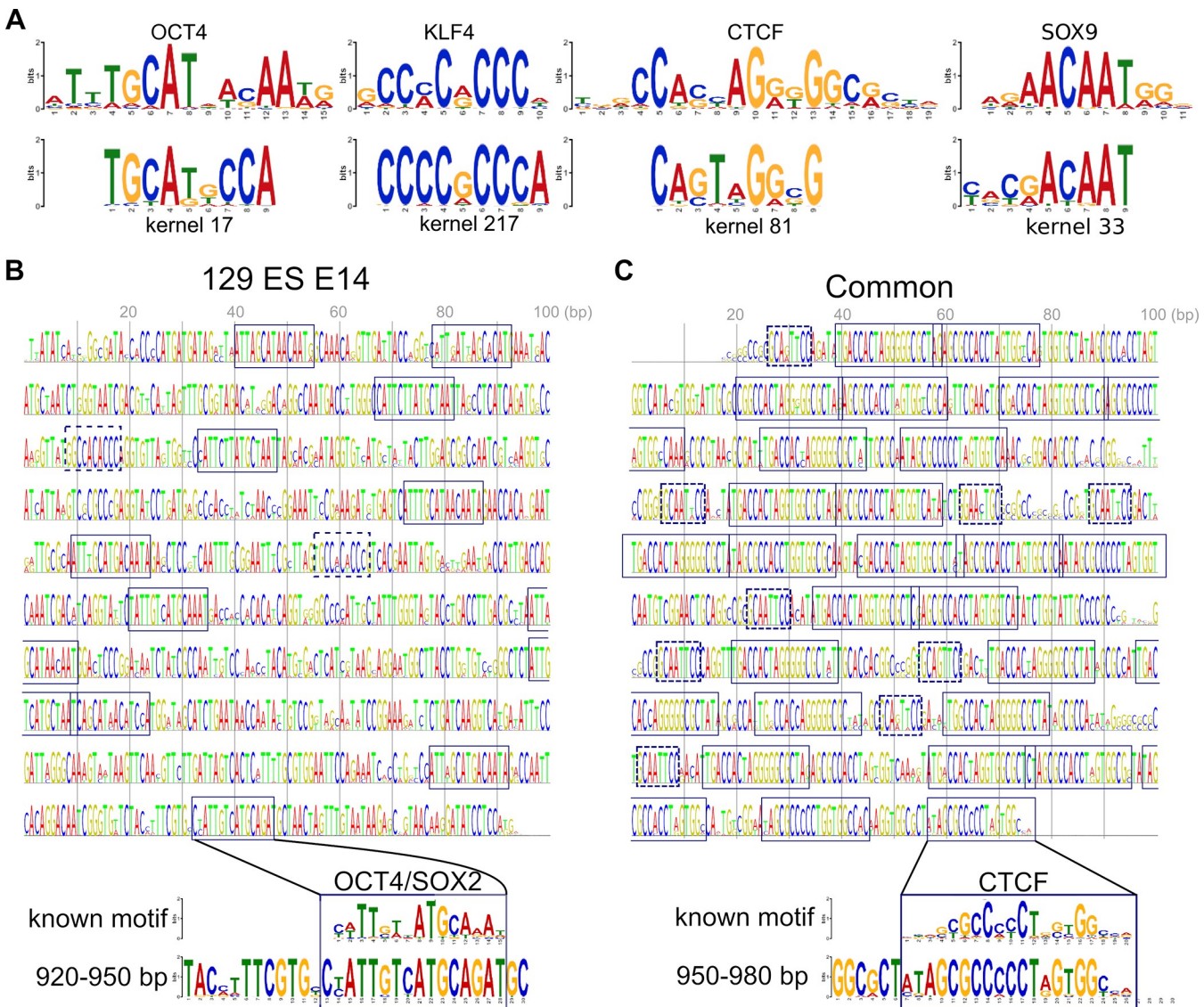

**Fig 4. Visualization of trained kernels and results of the activation maximization method.** (A) Examples of known motifs (top) that match with kernels. (B, C) Sequences trained to activate the 129 ES E14–specific neuron in the last layer (B) and all neurons in the last layer (C), respectively. Boxes in the left panel, OCT4/SOX2 binding motifs; dashed boxes in the left panel, KLF4 binding motifs; boxes in the right panel, CTCF binding motifs; dashed boxes, motifs similar to the second part of the alternative longer CTCF motif.

methods will provide researchers with clues for understanding the syntax of gene regulatory sequences.

We designed FRSS to process the information for forward and reverse sequences simultaneously, and we demonstrated the efficacy of FRSS for this purpose. This is the first to show that a new CNN architecture specialized for processing DNA sequences outperforms existing models. However, there are many ways for designing layers for this purpose. For example, instead of pair-wise summation in the last layer of FRSS, one can concatenate the output tensors similar to bidirectional RNN [32]. In addition, during finishing this study, we found a study that implemented a CNN model with a similar idea but a different architecture, although their model seemed to suffer from a run-time memory problem, and they did not clearly

compare the performance of their model with that of other models [33]. We choose the current FRSS architecture because its computational cost is smaller than that of other configurations. However, better architectures may be possible that retain low computational cost yet offer greater prediction power. Overall, our study indicates that neural net designs that take the nature of genomic molecules into account will yield high accuracy for genomic information processing. We released DeepGMAP (https://github.com/koonimaru/DeepGMAP), a software that allows users to train models and predict regulatory sequences in *de novo* genome sequences.

## Materials and methods

### Dataset and processing

The sequences for the human (hg19, hg38) and mouse (mm10) genomes were downloaded from UCSC (http://hgdownload.soe.ucsc.edu/goldenPath/). Mitochondrial DNA sequences were excluded from analyses. The genomic sequences were divided into 1-kbp windows with a variety of strides as described in Results. The filtered alignment files for epigenomic data were downloaded from the ENCODE website (https://www.encodeproject.org/; see S2 Table for details). A peak caller, MACS2 version 2.1.1.20160309, was used to determine signal peak regions with the following options: "callpeak—call-summits -t <target bam file> -c <control bam file> -f <BAM or BAMPE> -g <hs or mm> -q 0.01" for ChIP-seq, and "callpeak—call-summits -q 0.01 –nomodel–shift -100 –extsize 200 -t <target bam file> -f <BAM or BAMPE> -g <hs or mm>" for DNase-seq. Using the outputs of MACS2, bedtools [34], and our codes, genomic regions were designated as positive if windows were found to overlap with the summits of peaks (Fig 1A). DNase-seq data with < 10,000 peaks were excluded from training data because they showed an apparent poor quality. For comparative analyses in Fig 2A and 2B, peak files were downloaded from http://hgdownload.cse.ucsc.edu/goldenPath/hg19/encodeDCC/ (these peak files were generated by ENCODE using hg19). These data were used to mark genome sequences. For cross-species comparison, CanFam 3.1 was downloaded from NCBI (https://www.ncbi.nlm.nih.gov/genome/?term=dog), and the short reads for CTCF ChIP-seq data (ERR022304) were downloaded from EMBL-EBI ArrayExpress (https://www.ebi.ac.uk/arrayexpress/experiments/E-MTAB-437/).

We divided each of the mouse and human genome sequences into 1000-basepair (bp) windows and converted the four letters into one-hot vectors (four dimensional zero-one vectors). Namely, the DNA symbols, A, G, C, T and N were converted into one-hot vectors: (1, 0, 0, 0), (0, 1, 0, 0), (0, 0, 1, 0), (0, 0, 0, 1), and (0, 0, 0, 0), respectively. Therefore, one training sample became a 1000×4 tensor. To reduce the number of potential artifacts caused by window boundaries, we also added windows that were shifted by 500 bp toward the 3′ side (Fig 1A). With a mini-batch number of 100 for the stochastic gradient descent and channel dimension 1 (a dimension for colors if the input was image data), the final shape of the input tensor was 100×1000×4×1. To denote epigenomic marks, we assigned each window as 1 if it overlapped a signal positive region or 0 otherwise, which became a label tensor of size mini-batch number×-class number [i.e., 100 label vectors, and each label looks like (0, 0, 0, 1, 0, 1, . . .., 0); see the S1 Fig for visual illustration]. We generated several training datasets with different reference genome sequences (for example, we used the mm10 genome sequence for the mouse DNase-seq peaks that we determined with MACS2 and the hg19 genome sequence for the peak files downloaded from http://hgdownload.cse.ucsc.edu/goldenPath/hg19/encodeDCC/).

### Design of models and implementation

Tensorflow r1.8 for python (https://www.tensorflow.org/) with CUDA Driver v9.0 and cuDNN v7.0.5 was used as the main machine-learning library for model implementation. The

Tensorflow library was compiled from the source codes with the bazel compiler (https://bazel.build/; with options: "-c opt—copt = -mavx—copt = -mavx2—copt = -mfma—copt = -mfpmath = both—copt = -msse4.2—config = cuda -k //tensorflow/tools/pip_package:build_-pip_package"). For convolutions, we used a module, tensorflow.nn.conv2d, with options: "strides = [1, 1, 1, 1], padding = VALID". For the Basset model, because batch normalization did not yield any positive effects, we removed this operation. For RNN layers, we used the long short-term memory cells [35] (tensorflow.nn.rnn_cell.LSTMCell or tensorflow.contrib.rnn.LSTMBlockCell without the peephole option) and tensorflow.nn.bidirectional_dynamic_rnn for output calculations. The operations and hyperparameters of models are listed in S1 Table.

For the FRSS layers, convolution kernels were rotated 180 degrees to extract reverse complementary information in the first and second layers in parallel with normal convolutions as follows. Let $W^l$ a weight tensor of size M×N×H×K (M, kernel height; N, kernel width; H, input channel number; K, output channel number) in the $l$ th convolution layer, defined as $[W^l]_{m,n,h,k} = w^l_{m,n,h,k}$ ($[m] = \{0,\ldots,M-1\}; [n] = \{0,\ldots,N-1\}; [h] = \{0,\ldots,H-1\}; [k] = \{0,\ldots,K-1\}$). The rotated tensor $W^l_{rc}$ can be written as $[W^l_{rc}]_{m,n,h,k} = w^l_{(M-1-m),(N-1-n),h,k}$ (see the Fig 1C for a visual illustration for the rotation). Then, the first part of the FRSS layers is

$$X^1 = pool\Big(relu\big(conv(W^0, X^0)\big)\Big),$$

$$X^1_{rc} = pool\Big(relu\big(conv(W^0_{rc}, X^0)\big)\Big),$$

where $X^l$ is an input/output tensor of size B×S×N×H (B, mini batch number; S, sequence length; N, input width; H, input channel number) in the $l$ th convolution layer, $X^l_{rc}$ is the output of convolution with the rotated tensor, $conv$ is the convolution operation, $relu$ is the rectified linear unit, and $pool$ is the max pooling operation of size 2×1 and stride 2. The second part of the FRSS layers is

$$X^2 = pool\Big(relu\big(conv(W^1, X^1)\big) + relu\big(conv(W^1_{rc}, X^1)\big)\Big).$$

Note that the variables of $W^l_{rc}$ are not independent training targets but rather are the copies of those of $W^l$.

## Training and testing models

Stochastic gradient–based optimization was used for training models. For human data, the chromosome 8 and 9 were excluded from training data. For mouse data, the chromosome 2 was excluded. The excluded sequences were used to test trained models. hg19 genome was used for comparisons with previously reported AUPRCs (Fig 2A and 2B). A training dataset was subdivided into mini-batches containing 100 sequences. In the quick benchmark, because the fraction of positively labeled genomic windows was too small, negative samples were randomly downsampled to 25%. To accelerate the learning rate and compensate for the limited data size, each mini-batch was further divided into two sub-mini-batches, and updates were made alternately twice per sub-mini-batch (i.e., four updates per mini-batch). The loss function we used for training our models and the DeepSEA model is

$$\frac{1}{n}\sum_{k=0}^{n-1}(y_k log(\hat{y}_k) + (1 - y_k)log(1 - \hat{y}_k)) + \lambda_1 \|W\|_2^2 + \lambda_2 \|H\|_1,$$

where $y_k$ is a label of class $k$, $\hat{y}_k$ is a prediction after the sigmoid operation, $\|W\|_2^2$ is the L2 regularization of all variables, $\|H\|_1$ is the L1 regularization of predictions before the sigmoid

operation. $\lambda_1$ and $\lambda_2$ were set as $5 \times 10^{-7}$ and $1 \times 10^{-8}$, respectively according to [9]. For the remaining models, the loss function without the regularization terms was used. The optimization algorithms RMSprop [36] and Adam [37] were used to train the DanQ model and the others, respectively. To monitor training accuracy, models were tested by every mini-batch before updating variables and then evaluated based on the F1 score ($\frac{2 \cdot precision \cdot recall}{precision + recall}$) to assess temporal accuracy. When a mean of F1 scores of the last three tests exceeded a certain threshold (0.75), models were tested by three mini-batch sets that had been randomly excluded from a training dataset and saved when a higher F1 score was attained relative to the previous test.

NVIDIA TITAN X (Pascal; 3584 CUDA Cores; total amount of global memory, 12190 Mbytes; GPU Max Clock rate, 1531 MHz) was used for GPU-computing. Intel Xeon Processor E5-2640 v4 (2.40 GHz) was used for CPU-computing (see S1 Note for the full specification of our machine). Total training time was dependent on the number of classes, the amount of training data, and model architecture. For example, conv4-FRSS took three hours to train with the subset of mm10 DNase-seq data (three classes) and 11 hours with hg38 DNase-seq (365 classes). The bottleneck was, in part, to send tensor data from the GPU to CPU for evaluation of the temporal accuracy of the training models. In previous studies, training time was reported as 85 hours in the Basset paper with NVIDIA Tesla K20m and 164 classes, and as approximately 15 days in the DanQ paper with NVIDIA Titan Z and 919 classes. However, a fair comparison of training time with previous studies is difficult owing to differences in the machines, languages, and datasets.

To test trained models, the chromosome 2 sequence for mice and chromosome 8 and 9 sequences for humans were divided into the same window size and stride with those of the training data. AUROC and AUPRC were calculated with functions in the scikit-learn library version 0.19.1 (http://scikit-learn.org/stable/). We compared the accuracy between a model trained with the full training data and one saved during training, and adopted the one with higher AUPRCs. AUPRCs of DeepSEA and DanQ in the Fig 2A and 2B were obtained from S3 Table of the DeepSEA paper (https://media.nature.com/original/nature-assets/nmeth/journal/v12/n10/extref/nmeth.3547-S3.xlsx) and auc.txt of the DanQ repository (https://github.com/uci-cbcl/DanQ) and are listed in our S4 Table. Predictions presented in the Figs 1D and 3 were visualized using the integrative genomics viewer [38].

### The class saliency extraction method

To evaluate informative sequences at the single-nucleotide level (Fig 3 and S5 Fig), the class saliency extraction method was performed as previously described [25] with slight modifications. Let $S_i(X^0)$ be the score of the class $i$ before sigmoidal operation, computed by the conv4-FRSS model for an arbitrary DNA sequence $X^0$ of size $1000 \times 4 \times 1$. In other words, $S_i(X^0)$ is a function that includes all operations such as FRSS, convolution and fully-connected layers except the last sigmoidal operation. The derivative of $S_i(X^0)$ with respect to $X^0$ is

$$\omega = \frac{\partial S_i(X^0)}{\partial X^0}\bigg|_{X_0^0},$$

where $X_0^0$ is a given DNA sequence. The derivative $\omega$ is a tensor of size $1000 \times 4 \times 1$, and each element corresponds to each element of $X^0$. The values shown in the Fig 3A and S5 Fig (orange) are $\sum_n |\omega_{s,n,h}|$ ($s = 0, \ldots, 999$; $n = 0, \ldots, 3$; $h = 0$). To remove irrelevant values, the figures only show sequences with $S_i(X^0) \geq -2.0$. These values represent the magnitude of the effect on the class score when nucleotide substitutions occur. Therefore, these values may be termed as influential values.

### FRiP calculation and correction

To calculate FRiPs, initially, mitochondrial and black-listed regions (https://sites.google.com/site/anshulkundaje/projects/blacklists) were removed from peak files. A module, "countReadsPerBin.CountReadsPerBin" in deeptools [39], was used to count reads in peaks, and these read counts were then divided by total reads. As a correction for read number differences, FRiP was multiplied by the fraction of genome regions covered by at least one read.

### Visualizing the inside of a trained model

Variables in each kernel were converted to a probability matrix by a softmax-like operation. Tomtom [29] version 4.12.0 and MEME motif database version 12.15 (http://meme-suite.org/meme-software/Databases/motifs/motif_databases.12.15.tgz) were used to identify closely related known transcription factor binding motifs. For visualizing kernels in the first layer of FRSS, the probability matrices were scaled by information content. Motif logos except Tomtom outputs were generated by our customized codes using the cairocffi library (https://cairocffi.readthedocs.io/en/latest/).

### The activation maximization method

Let $\hat{y}_i(X^0)$ be the score of the class $i$ ($i = 0,1,2$), computed by the conv4-FRSS model for a sequence $X^0$. To find a sequence that is specific to class $i$, we sought to optimize the following problem:

$$\arg\max_{X^0} \frac{\hat{y}_i(X^0)}{\sum_{k \neq i}\hat{y}_k(X^0)} - \lambda\|X^0\|_2^2,$$

where $\lambda = 5.0 \times 10^{-3}$. For the common sequence (right panel in the Fig 4), the problem was:

$$\arg\max_{X^0} \prod \hat{y}_i(X^0) - \lambda\|X^0\|_2^2.$$

These formulae differ slightly from those reported previously [25] because our model used the sigmoid function for the last output, whereas the others used softmax. The optimizer Adam was used to optimize the target sequence. The initialization of variables of $X^0$ was set to have a mean of 0.02 and stddev of 0.02, and this was a critical factor for this optimization problem. Training with different initialization conditions sometimes did not converge, suggesting that the optimization landscape is rugged and there is a possibility that the results shown in the Fig 4 are not the optimal solution. The generated sequences were visualized with our customized codes using the cairocffi library. Tomtom was used to find motifs in the generated sequences.

## Supporting information

**S1 Fig. Illustration of how a convolutional neural network reads DNA sequences.** (**A**) One-hot vectors that represent a DNA sequence. A 1-kbp DNA sequence is converted into a 1000×4 matrix. (**B**) A general convolutional network for sequence classifications. Only the first layer is shown, as an example. In the first layer, 9×4 kernels "scan" a sequence by striding base pair-by-base pair. Each "scan" is actually an element-wise multiplication of the values of a kernel (wmnhk) with values of 0 or 1 in a part of the sequence, yielding a scalar value represented by the black-to-white colored boxes shown below. By striding, scalar values are concatenated to a vector of size 992 (1000–9 + 1). Because the layer has multiple kernels (320 in this case), the output of the layer is 320 vectors. Maxpooling chooses maximum elements in each window. In this case, because the window size is 2, the total length of the sequence is halved by

pooling. After fully connecting the layers, sigmoid operations yield output values between 0 and 1, which is compared with the class labels of "0" and "1".
(PDF)

**S2 Fig. AUPRCs of the DeepSEA model.** Boxplots of AUPRCs sorted by factors. The data were derived from nmeth.3547-S3.xlsx of Zhou & Troyanskaya (2015). Boxes, the lower-to-upper quantile values for the data; orange lines, the median; whiskers, the range of the data. Flier points, those past the end of the whiskers. Note that the prediction of CTCF binding sites (red rectangle) had the best performance among transcription factors.
(PDF)

**S3 Fig. Scatter plots for FRiP and corrected FRiP with AUPRCs (top panels) and peak numbers (bottom panels).** AUPRCs are the prediction accuracy of conv4-FRSS trained with mm10 CTCF dataset. Black arrows indicate data with low AUPRCs but relatively high FRiP scores. The quality of these data is reasonably estimated with cFRiP. The validity of cFRiP is also supported by the better correlation with peak numbers. corr., Spearman's correlation.
(PDF)

**S4 Fig. The average of AUPRCs with various cutoff values of corrected FRiP.** (**A, B**) Plots of average AUPRCs for human and mouse DNase-seq data (**A** and **B**) and CTCF data (**C** and **D**) as a function of cutoff values of corrected FRiPs. Note that increase in cutoff values of the corrected FRiPs results in an increase of the average AUPRCs, at least to some extent, except as shown in **c**. Red line, cutoff values for the filtered data in Fig 2A and 2B.
(PDF)

**S5 Fig. Evaluation of informative sequences at the single-nucleotide level.** (**A**) CTCF binding distribution of mouse liver E14.5, as an example. Influential value, the values calculated by the class saliency extraction method; Prediction, CTCF binding sites predicted with conv4-FRSS; CTCF signal, CTCF ChIP-seq signals derived from ENCFF844ZSH; red rectangle, a region magnified in b. Note that the distribution of influential values is similar to that of the CTCF signal although the model is not directly trained with the signal data. (**B**) Comparison of influential values and CTCF binding motif. The CTCF binding motifs were detected with FIMO. Note that nucleotides with high influential value (orange) are aligned with the high information content sites in the CTCF motif (arrows).
(PDF)

**S6 Fig. All kernels of conv4-FRSS trained by a subset of the mouse DNase-seq data.** Note that several kernels contain little information and thus appear empty. These kernels may suggest either that the kernel number was sufficiently set or that training was insufficient to optimize all kernels.
(PDF)

**S1 Note.**
(PDF)

**S1 Table.**
(XLSX)

**S2 Table.**
(XLSX)

**S3 Table.**
(XLSX)

**S4 Table.**
(XLSX)

**S5 Table.**
(XLSX)

**S6 Table.**
(XLSX)

**S7 Table.**
(XLSX)

**S8 Table.**
(XLSX)

# Acknowledgments

We thank Dr. Yuichiro Hara for critical comments and Dr. Mitsutaka Kadota for discussions concerning unpublished data.

# Author Contributions

**Conceptualization:** Koh Onimaru.

**Data curation:** Koh Onimaru.

**Formal analysis:** Koh Onimaru.

**Funding acquisition:** Koh Onimaru, Shigehiro Kuraku.

**Investigation:** Koh Onimaru.

**Methodology:** Koh Onimaru, Osamu Nishimura.

**Project administration:** Koh Onimaru.

**Resources:** Koh Onimaru.

**Software:** Koh Onimaru.

**Supervision:** Shigehiro Kuraku.

**Validation:** Koh Onimaru.

**Visualization:** Koh Onimaru.

**Writing – original draft:** Koh Onimaru.

**Writing – review & editing:** Koh Onimaru, Osamu Nishimura, Shigehiro Kuraku.

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
