## [Decision Letter · Decision Letter 0]

16 Mar 2020

PONE-D-19-33090

Predicting gene regulatory regions with a convolutional neural network for processing double-strand genome sequence information

PLOS ONE

Dear Onimaru,

Thank you for submitting your manuscript to PLOS ONE. After careful consideration, we feel that it has merit but does not fully meet PLOS ONE’s publication criteria as it currently stands. Therefore, we invite you to submit a revised version of the manuscript that addresses the points raised during the review process.

We would appreciate receiving your revised manuscript by Apr 30 2020 11:59PM. To enhance the reproducibility of your results, we recommend that if applicable you deposit your laboratory protocols in protocols.io, where a protocol can be assigned its own identifier (DOI) such that it can be cited independently in the future. For instructions see: http://journals.plos.org/plosone/s/submission-guidelines#loc-laboratory-protocols

We look forward to receiving your revised manuscript.

Kind regards,

Jiajie Peng

Academic Editor

PLOS ONE

Journal Requirements:

Reviewers' comments:

Reviewer's Responses to Questions

**Comments to the Author**

1. Is the manuscript technically sound, and do the data support the conclusions?

Reviewer #1: Partly

2. Has the statistical analysis been performed appropriately and rigorously? 

Reviewer #1: No

3. Have the authors made all data underlying the findings in their manuscript fully available?

Reviewer #1: Yes

4. Is the manuscript presented in an intelligible fashion and written in standard English?

Reviewer #1: No

5. Review Comments to the Author

Reviewer #1: Annotating biological functions in genome sequences without experiments is still a challenging task. In this manuscript, a deep-learning-based method was proposed to predict gene regulatory regions efficiently.

I have the following major concerns.

Fig 1 is blurry.

Line 71, page 4, references are missing for DeepSEA and Basset.

Line 274, page 16, what are the meanings of labels “0” and “1”?

Line 293, page 17, more explanations are needed for the rotated kernel. Some examples of the kernel may help understand this concept.

Line 298, page 17, what is the max pooling size?

Line 263, page 15, “Data with < 10,000 peaks were excluded from training data.” Why is that?

Line 264, page 15, “peak files were downloaded from http://hgdownload.cse.ucsc.edu/goldenPath/hg19/encodeDCC/.” Are these peak files matching with hg38? What about the peak files for mouse genomes?

Line 313, page 18, “λ1 = 5×10–7, and λ2 = 1×10–8” More explanations are needed for setting these two parameters.

Line 346, page 19, what is the formula for Si(X0)?

6. PLOS authors have the option to publish the peer review history of their article (what does this mean?). If published, this will include your full peer review and any attached files.

Reviewer #1: No

---

## [Author Response · Author response to Decision Letter 0]

3 Apr 2020

We thank the reviewer for his/her valuable comments. Indeed, we agree that the points raised were unclear, and we have therefore used this input to make improvements to the manuscript. 

Reviewers' comments:

Reviewer's Responses to Questions

Comments to the Author

======Comments======

Reviewer #1: Annotating biological functions in genome sequences without experiments is still a challenging task. In this manuscript, a deep-learning-based method was proposed to predict gene regulatory regions efficiently.

I have the following major concerns.

Fig 1 is blurry.

=======Response======

We apologize for the problem. Because all figures were created with a same resolution, we have no idea what went wrong with Figure 1. At least, the resolution of Figures in the submitted manuscript was fine on my computer. We have double-checked the resolution of all figures. As the next submission will contain the figures as individual tiff files, the resolution will probably be better than the first submission.

======Comments======

Line 71, page 4, references are missing for DeepSEA and Basset.

=======Response======

We have added references.

======Comments======

Line 274, page 16, what are the meanings of labels “0” and “1”?

=======Response======

We agree that the explanation for generating training data was not clear enough. We assigned each window as 1 if an input sequence overlapped a signal positive region or 0 otherwise. We have improved the explanation for generating the training data (lines 287–298).

======Comments======

Line 293, page 17, more explanations are needed for the rotated kernel. Some examples of the kernel may help understand this concept.

=======Response======

We agree that we should have make the explanation for the rotated kernel easier to understand because this is the key concept of this paper. There was a visual explanation for this rotation in Figure 1C. Now, we have modified Figure 1C to be more consistent with the mathematical notation and cited Figure 1C in the sentence (Figure 1C and line 323). 

======Comments======

Line 298, page 17, what is the max pooling size?

=======Response======

We have added the size of the max pooling (line 328). 

======Comments======

Line 263, page 15, “Data with < 10,000 peaks were excluded from training data.” Why is that?

=======Response======

We apologize for the poor explanation. First of all, one thing that we did not mention here was that this filtering was applied only for DNase-seq data. We filtered out DNase-seq data with a few peaks because they usually showed a poor quality and disrupted the training. We have added this explanation in lines 278–279.

======Comments======

Line 264, page 15, “peak files were downloaded from http://hgdownload.cse.ucsc.” Are these peak files matching with hg38? What about the peak files for mouse genomes?

=======Response======

We agree that the explanation for generating training data was not clear enough. The reference of the peak files downloaded from the UCSC is hg19. These data are used for comparing the performance of conv4-frss, DeepSEA and DanQ, because DeepSEA and DanQ used these downloaded peaks for training (DeepSEA, DandQ and conv4-FRSS in Figure 2A and B). Separately, to utilize newly added data by the ENCODE project, which uses the mm10 and hg38 genomes, we created our own training datasets for other analyses such as Table 1 and “hg38” and “mm10” in Figure 2A and B). We have modified the manuscript to make that clear (lines 281–282 and lines 298–301).

======Comments======

Line 313, page 18, “λ1 = 5×10–7, and λ2 = 1×10–8” More explanations are needed for setting these two parameters.

=======Response======

These two parameters are set according to a previous paper (Zhou et al 2015). We have added this explanation to the manuscript (line 348).

======Comments======

Line 346, page 19, what is the formula for Si(X0)?

=======Response======

The formula for Si(X0) is too complex to write here, because it is composed of all operations including FRSS, convolution and fully-connected layers. We have added this explanation to the manuscript (lines 386–388).

---

## [Decision Letter · Decision Letter 1]

23 Jun 2020

Predicting gene regulatory regions with a convolutional neural network for processing double-strand genome sequence information

PONE-D-19-33090R1

Dear Dr. Onimaru,

We’re pleased to inform you that your manuscript has been judged scientifically suitable for publication and will be formally accepted for publication once it meets all outstanding technical requirements.

Kind regards,

Vladimir Makarenkov

Academic Editor

PLOS ONE

Additional Editor Comments (optional):

Reviewers' comments:

Reviewer's Responses to Questions

**Comments to the Author**

1. If the authors have adequately addressed your comments raised in a previous round of review and you feel that this manuscript is now acceptable for publication, you may indicate that here to bypass the “Comments to the Author” section, enter your conflict of interest statement in the “Confidential to Editor” section, and submit your "Accept" recommendation.

Reviewer #1: All comments have been addressed

Reviewer #2: All comments have been addressed

2. Is the manuscript technically sound, and do the data support the conclusions?

Reviewer #1: Yes

Reviewer #2: Partly

3. Has the statistical analysis been performed appropriately and rigorously? 

Reviewer #1: Yes

Reviewer #2: N/A

4. Have the authors made all data underlying the findings in their manuscript fully available?

Reviewer #1: Yes

Reviewer #2: Yes

5. Is the manuscript presented in an intelligible fashion and written in standard English?

Reviewer #1: Yes

Reviewer #2: Yes

6. Review Comments to the Author

Reviewer #1: (No Response)

Reviewer #2: The authors have carefully addressed given comments raised in the previous round of review by other reviewers. Hence, I think this version can be accepted in this round as is.

7. PLOS authors have the option to publish the peer review history of their article (what does this mean?). If published, this will include your full peer review and any attached files.

Reviewer #1: No

Reviewer #2: No

---

## [Editor Report · Acceptance letter]

1 Jul 2020

PONE-D-19-33090R1 

Predicting gene regulatory regions with a convolutional neural network for processing double-strand genome sequence information 

Dear Dr. Onimaru:

I'm pleased to inform you that your manuscript has been deemed suitable for publication in PLOS ONE. Congratulations! Your manuscript is now with our production department. 

Kind regards, 

on behalf of

Dr. Vladimir Makarenkov 

Academic Editor

PLOS ONE